# STRUCT-G: STRUCTURAL-AWARE PRETRAINING FOR GRAPH AND TASK TRANSFER LEARNING

## ABSTRACT

Transfer learning has revolutionized domains like vision and language by enabling pretrained models to adapt rapidly with minimal supervision. However, applying transfer learning to graph-structured data faces unique challenges: graphs exhibit diverse topology, sparse or heterogeneous node attributes, and lack consistent semantics across datasets, making it difficult to learn representations that generalize across domains. Recent graph pretraining efforts including generative methods and contrastive objectives have shown promise but often rely on complex architectures, rich feature modalities, or heavy computation, limiting their applicability to structure-only graphs and resource-constrained settings. In an attempt to address these challenges, we introduce STRUCT-G, a lightweight pretraining model that decouples global topology capture from local feature refinement. STRUCT-G first computes shallow random-walk–based structural embeddings, then fuses them with raw attributes via an adaptive, feature-wise gating network and a shared message-passing backbone. By jointly optimizing multiple self-supervised objectives such as link prediction, node classification, feature reconstruction, and structural alignment, STRUCT-G learns robust node embeddings that transfer effectively with fine-tuning. Our extensive experiment results demonstrate that explicit structural inductive bias and self-supervised multi-task learning provide a scalable and accessible foundation for graph representation learning.

## 1 INTRODUCTION

Transfer learning (Zhuang et al., 2020) is an important topic in modern machine learning. In computer vision and natural language processing domains, transfer learning reduces the data and computational requirements of downstream tasks. Graph domains, however, present unique obstacles: irregular connectivity patterns, sparse or heterogeneous node attributes, and varying semantics across datasets make it difficult for a graph neural network (GNN) trained on one task or domain to generalize to another (Lee et al., 2017; Xu et al., 2023). As a result, standard GNNs often require extensive retraining or large labeled cohorts to perform well on new graphs, undermining the efficiency benefits of transfer learning.

To overcome these challenges, recent studies have explored graph transfer learning through self-supervised pretraining, where models are first trained on large unlabeled graphs using carefully designed pretext tasks and later adapted to downstream tasks with limited supervision. Generative approaches (Li et al., 2023; Hu et al., 2020; Sun et al., 2022) mask and reconstruct node features or substructures, teaching the model to capture local and mid-range patterns. Contrastive methods (You et al., 2020; Sun et al., 2019) create augmented views of the same graph—through edge perturbations, node dropping, or subgraph sampling—and train the encoder to bring these views' embeddings closer while pushing apart different graphs. More recently, prompting-based techniques (Liu et al., 2023) have reframed downstream tasks as learned queries to a pretrained graph encoder, offering a flexible way to adapt the foundation model.

Contributing to this area of research, we present STRUCT-G, a novel graph pretraining model designed to capture and transfer global structural patterns in various graph settings. Our key contributions are: (i) a lightweight structural encoding step based on random-walk embeddings, which precomputes high-order topology in a compact form and remains frozen during downstream training to provide stable global context; (ii) an adaptive, feature-wise gating mechanism that fuses

structural encodings with raw node attributes, allowing the model to dynamically balance topology and features on a per-node and per-task basis; and (iii) a unified multi-task self-supervised pretraining objective—comprising link prediction, node classification, feature reconstruction, and structural alignment—that instills robust, transferable invariances in the shared graph encoder. Our extensive experiment results demonstrate that STRUCT-G achieves faster convergence, superior performance compared to the existing methods.

## 2 RELATED WORK

Pre-training has proven transformative in language and vision domains, and recent years have seen substantial effort to replicate this success in graph representation learning. Graph pre-training methods typically aim to learn transferable structural and semantic representations by solving self-supervised tasks before downstream fine-tuning. These methods differ in both pretext task design and the level of supervision—node-level, edge-level, or graph-level.

Early graph autoencoders and masked-feature models focus on reconstructing hidden inputs. Hu et al. (Hu et al., 2019) introduce attribute masking, context prediction, and graph-level classification tasks, demonstrating transferability across multiple datasets. Lu et al. (Lu et al., 2021) further analyze the effect of pre-training on different levels of graph granularity. Jiang et al. (Jiang et al., 2021) extend this to heterogeneous graphs, while Wang et al. (Wang et al., 2021) explore pre-training in the cross-domain recommendation setting. These works emphasize the importance of pretext-task design and alignment with downstream objectives.

Another class of methods adapts generative and autoregressive objectives. GPT-GNN (Hu et al., 2020) introduces a framework that pre-trains GNNs via edge sequence generation, drawing analogies to language modeling. GROVER (Rong et al., 2020) employs masked subgraph and motif prediction to pre-train transformers on molecular graphs, achieving state-of-the-art property prediction. GPPT (Sun et al., 2022) combines pre-training with prompt tuning to reduce the need for task-specific supervision. Similarly, GraphPrompt (Liu et al., 2023) introduces a unified view of graph pre-training and prompting, enabling task adaptation through learned queries rather than fine-tuning the full model.

Contrastive objectives have gained popularity in self-supervised graph learning. GCC (Qiu et al., 2020) formulates graph contrastive coding by aligning node representations across graph augmentations at multiple scales. This approach promotes generalization without explicit labels, and has inspired a range of subsequent models in the contrastive graph learning literature. GraphCL (You et al., 2020) maximizes mutual information between global and local views or between distinct graph augmentations.

Inspired by large language models (LLMs), researchers have begun incorporating graph structure into language-based pre-training pipelines. Xie et al. (Xie et al., 2023) propose a graph-aware pre-training framework on large text–graph corpora to unify graph learning and language modeling. These approaches blur the boundary between graph and textual modalities and open new avenues for multi-modal graph tasks.

Recent work has questioned the universality of pre-training benefits. Cao et al. (Cao et al., 2023) investigate the conditions under which pre-training improves downstream performance, highlighting the influence of data distributions and the structure-task gap. Similarly, Min et al. (Min et al., 2022) explore the mismatch between graph structure and recommendation-specific semantics, suggesting that pretext objectives must be carefully aligned with target domains.

We position our work at the intersection of structural embedding and GNN-based pre-training. Instead of relying solely on attribute- or topology-based SSL objectives, we incorporate explicit random-walk-based embeddings into a unified GNN architecture. Our model is trained end-to-end across multiple tasks.

## 3 PROBLEM STATEMENT

We consider the setting of transfer learning on graphs. Formally, let $G = (\mathcal{V}, \mathcal{E})$ denote a graph with $N = |\mathcal{V}|$ nodes, feature matrix $\mathbf{X} \in \mathbb{R}^{N \times d_{\text{in}}}$, and a small labeled subset $\mathcal{L} \subset \mathcal{V}$, where $|\mathcal{L}| \ll N$.

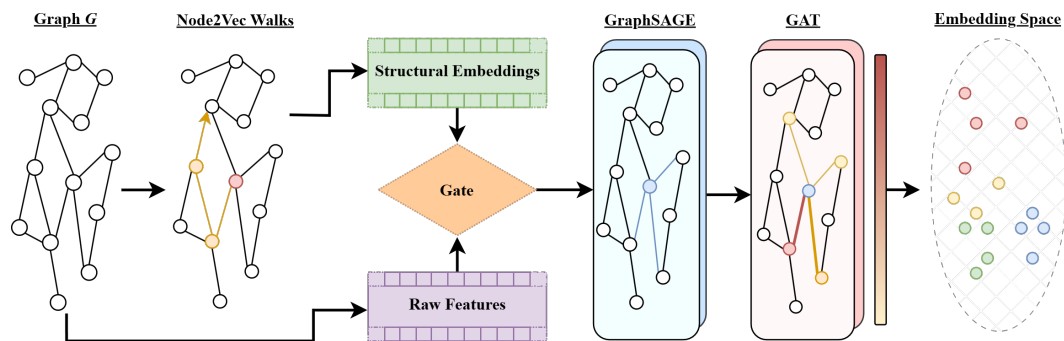

Figure 1: STRUCT-G framework. Raw node features and structural signatures (from random walk embeddings) are fused via a learnable gate. The resulting embeddings are passed through a shared GraphSAGE-based encoder with GAT to produce a unified node representation used across multiple tasks.

Our goal is to learn an encoder $f : \mathcal{V} \to \mathbb{R}^{d_{\text{out}}}$ that produces node embeddings $\mathbf{e}_v = f(v)$ which can be efficiently adapted to multiple downstream tasks such as node classification, link prediction, and attribute reconstruction. The key requirement is that $f$ should enable effective transfer with limited supervision and modest computational overhead.

This problem is challenging because real-world graphs vary widely in connectivity patterns (e.g., homophily vs. heterophily, strong clustering vs. scale-free structure). Moreover, labeled data is often scarce, and practical applications require models that scale to large graphs without prohibitive training cost.

## 4 STRUCT-G MODEL

We propose STRUCT-G, a unified pretraining framework that separates global topology capture from local feature learning without relying on rich node attributes or complex generative models. Figure 1 illustrates the overall framework of our method. Each node is simultaneously described by a structural signature that is derived from random walk statistics to capture high-order connectivity, and a lightweight, trainable GNN encoder that propagates and refines raw attributes via message-passing. These two streams are merged through a learnable, element-wise gating module that dynamically adjusts the influence of structural versus feature-based signals. During pretraining, STRUCT-G optimizes a combination of self-supervised tasks to ensure that the resulting embeddings encode diverse, transferable invariants.

### 4.1 STRUCTURAL SIGNATURE

To generate robust and transferable graph representations, STRUCT-G first computes a *structural signature* for each node by obtaining a shallow random-walk embedding model (e.g., Node2Vec grover2016node2vec) on the entire graph, thereby capturing high-order topology. Understanding such topology is crucial since many graph phenomena like community structures, structural roles, and long-range dependencies span multiple hops and are beyond the reach of purely local aggregation. Specifically, we use biased random walks over $G = (\mathcal{V}, \mathcal{E})$ and treat the resulting node sequences as a corpus. We then optimize the skip-gram objective with negative sampling as follows:

$$\max_{\Theta} \sum_{u \in \mathcal{V}} \sum_{v \in \mathcal{C}_k(u)} \left[ \log \sigma\big(z_u^{(0)\top} z_v^{(0)}\big) \; + \sum_{v' \sim P_{\text{neg}}} \log \sigma\big(-z_u^{(0)\top} z_{v'}^{(0)}\big) \right],$$

where $z_u^{(0)} \in \mathbb{R}^{d_s}$ is the learned structural vector for node $u$, $\mathcal{C}_k(u)$ its context window of size $k$, and $P_{\text{neg}}$ the negative-sampling distribution. After training, we assemble these into a fixed matrix $Z^{(0)} = \big[\, z_u^{(0)} \,\big]_{u \in \mathcal{V}}$ and freeze it for all subsequent GNN training. By offloading high-order pattern extraction to this lightweight embedding step, our main encoder can dedicate its capacity to refin-

ing and aligning these global cues with node features, resulting in faster convergence and stronger transferability.

## 4.2 ADAPTIVE GATING FOR STRUCTURAL–FEATURE FUSION

To effectively combine global structural priors with local node features, STRUCT-G introduces a learnable, element-wise fusion gate that modulates the contribution of each modality on a per-node basis. This mechanism is particularly valuable in multi-task scenarios: for instance, when link prediction benefits from high-order structural cues, while attribute imputation depends more on raw features. Our gating module allows the network to dynamically interpolate between these signals, adapting to the demands of the task and the quality of the inputs.

Let $z_v^{(0)} \in \mathbb{R}^{d_s}$ denote the frozen structural signature obtained via Node2Vec, and $x_v \in \mathbb{R}^{d_x}$ the raw feature vector for node $v$. We first project $z_v^{(0)}$ into the GNN's hidden space:

$$\tilde{z}_v = W_{\text{proj}} \, z_v^{(0)}, \qquad \tilde{z}_v \in \mathbb{R}^{d_h}.$$

We then concatenate this with the raw features to form a fused input vector:

$$\nu_v = [\, x_v \,\|\, \tilde{z}_v \,] \in \mathbb{R}^{d_x + d_h}.$$

A learnable gate $g_v \in (0,1)^{d_h}$ is produced via a sigmoid-activated linear transformation:

$$g_v = \sigma(W_g \, \nu_v + b_g).$$

This gate modulates two candidate hidden states, both residing in the same latent space via a shared projection matrix $W_c$:

$$\hat{h}_v = W_c \, \nu_v, \qquad \check{h}_v = W_c \, [\, x_v \,\|\, \mathbf{0} \,],$$

where $\mathbf{0}$ is a zero vector of length $d_h$. The resulting fused representation is a convex combination:

$$h_v^{(0)} = g_v \odot \hat{h}_v + (1 - g_v) \odot \check{h}_v,$$

where $\odot$ denotes element-wise multiplication.

By adjusting $g_v$ at the level of individual nodes and dimensions, the model can suppress unreliable features, downweight uninformative structural priors, or interpolate adaptively between the two. This flexible gating enables more robust initialization and improves transferability across diverse downstream tasks.

## 4.3 UNIFIED GRAPH ENCODING BACKBONE

To support multiple self-supervised objectives with a shared representation, STRUCT-G employs a single, parameter-shared message-passing stack that produces a unified node embedding for all downstream tasks. This shared backbone ensures that improvements made by optimizing one task (e.g., link prediction) can transfer to others (e.g., node classification or feature reconstruction), rather than being confined to siloed branches.

The core architecture comprises a sequence of GraphSAGE layers using the mean aggregator hamilton2017inductive, optionally followed by an attention-based refinement step, and finalized with a projection to a universal embedding space:

$$\underbrace{\text{SAGE}_1 \circ \cdots \circ \text{SAGE}_L}_{\text{Backbone layers}} \xrightarrow{\text{(optional)}} \underbrace{\text{GAT}}_{\text{attention refinement}} \rightarrow \underbrace{\text{SAGE}_{\text{out}}}_{d_h \rightarrow d_{\text{out}}}.$$

Starting from the fused node representations $h_v^{(0)}$ (see Section 4.2), each GraphSAGE layer applies localized neighborhood aggregation:

$$h_v^{(\ell+1)} = \sigma \left( W^{(\ell)} h_v^{(\ell)} + U^{(\ell)} \sum_{u \in \mathcal{N}(v)} h_u^{(\ell)} \right), \qquad \ell = 0, \dots, L-1,$$

where $\sigma$ denotes ReLU, and the mean aggregator is normalized internally as per `pyg.nn.SAGEConv`. This stack progressively refines the node states while preserving computational efficiency.

To further sharpen neighborhood influence, we optionally insert a two-head GAT layer:

$$h_v^\star = \sigma\left(\text{GATConv}(h^{(L)}, \mathcal{E})_v\right).$$

This attention-based refinement allows the model to reweight neighbors dynamically based on learned edge-level importance. If disabled, we simply take $h_v^\star = h_v^{(L)}$.

Finally, a projection layer maps the hidden state into a universal embedding space of dimension $d_{\text{out}}$ (default: 32):

$$e_v = \text{SAGE}_{\text{out}}(h^\star, \mathcal{E})_v.$$

The resulting vector $e_v$ is the shared representation fed to all self-supervised heads, promoting alignment across objectives and enhancing generalization without requiring task-specific branches.

## 4.4 TASK-SPECIFIC HEADS AND MULTI-OBJECTIVE PRETRAINING

To maximize generality and transferability, STRUCT-G jointly optimizes a suite of self-supervised and supervised tasks over a shared embedding space. Each task operates on the backbone embeddings $\mathbf{e}_v \in \mathbb{R}^{d_{\text{out}}}$ and contributes to a composite training objective. Crucially, all task-specific heads share parameters wherever possible to encourage feature reuse and cross-task regularization.

Given ground-truth labels $y_v \in \{1, \ldots, C\}$ for nodes in a training subset $\mathcal{V}_{\text{train}}$, we apply a linear classifier:

$$\hat{\mathbf{y}}_v = W_{\text{cls}}\mathbf{e}_v + \mathbf{b}_{\text{cls}}, \quad W_{\text{cls}} \in \mathbb{R}^{C \times d_{\text{out}}}.$$

The associated loss is standard cross-entropy:

$$\mathcal{L}_{\text{cls}} = \frac{1}{|\mathcal{V}_{\text{train}}|} \sum_{v \in \mathcal{V}_{\text{train}}} H\left(\hat{\mathbf{y}}_v, y_v\right).$$

To capture structural connectivity, we predict edges by contrasting true edges against random negative samples. Each node's embedding is concatenated with its frozen structural signature:

$$\mathbf{u} = [\mathbf{e}_u \,\|\, \mathbf{z}_u^{(0)}], \quad \mathbf{v} = [\mathbf{e}_v \,\|\, \mathbf{z}_v^{(0)}],$$

and passed through a multi-layer perceptron over the pairwise triple $[\mathbf{u} \,\|\, \mathbf{v} \,\|\, \mathbf{u} \odot \mathbf{v}]$. The link prediction loss is binary cross-entropy over positive edges $(u, v) \in \mathcal{E}$ and $K$ negative samples:

$$\mathcal{L}_{\text{lp}} = \mathbb{E}_{(u,v) \sim \mathcal{E}} \left[ \text{BCE}(s_{uv}, 1) + \sum_{k=1}^{K} \text{BCE}(s_{uv_k^-}, 0) \right].$$

To ensure that embeddings retain semantic information from input attributes, we optionally decode them via a lightweight MLP $f_{\text{dec}} : \mathbb{R}^{d_{\text{out}}} \to \mathbb{R}^{d_x}$:

$$\hat{\mathbf{x}}_v = f_{\text{dec}}(\mathbf{e}_v), \quad \mathcal{L}_{\text{rec}} = \frac{1}{|V|} \sum_{v \in V} \|\hat{\mathbf{x}}_v - \mathbf{x}_v\|_2^2.$$

To preserve structural priors in the learned embeddings, we encourage alignment between the backbone output and the original Node2Vec signatures. This is done by projecting both vectors into a common space via learnable MLPs $\phi$ and $\psi$, followed by cosine similarity after $\ell_2$ normalization:

$$\mathcal{L}_{\text{align}} = \frac{1}{|V|} \sum_{v \in V} \left( 1 - \left\langle \frac{\phi(\mathbf{e}_v)}{\|\phi(\mathbf{e}_v)\|_2}, \frac{\psi(\mathbf{z}_v^{(0)})}{\|\psi(\mathbf{z}_v^{(0)})\|_2} \right\rangle \right).$$

This auxiliary objective nudges the encoder to retain global structure even as it adapts to downstream tasks.

All active losses are combined into a single training signal:

$$\mathcal{L}_{\text{total}} = \lambda_{\text{cls}}\mathcal{L}_{\text{cls}} + \lambda_{\text{lp}}\mathcal{L}_{\text{lp}} + \lambda_{\text{rec}}\mathcal{L}_{\text{rec}} + \lambda_{\text{align}}\mathcal{L}_{\text{align}},$$

where weights $\lambda_\bullet$ control task importance and can be toggled to activate or suppress individual components. This multitask design allows STRUCT-G to integrate supervision signals from multiple modalities—topology, attributes, labels—without requiring bespoke encoders for each.

## 5 EXPERIMENTAL FRAMEWORK

To evaluate the effectiveness and generality of STRUCT-G, we design a systematic experimental framework aligned with three key research questions:

**RQ1: Performance on Downstream Tasks.** *Does* STRUCT-G *improve predictive performance on downstream tasks compared to baselines across diverse real-world graphs?*

**RQ2: Transfer Learning Performance.** *How does* STRUCT-G *perform on inter-dataset tasks?*

**RQ3: Objective-Level Contributions.** *What are the individual contributions of each self-supervised objective to downstream performance?*

### 5.1 DATASETS AND GRAPH DIVERSITY

To evaluate generalization and efficiency across a range of domains and graph scales, we benchmark all models on five real-world networks spanning diverse application areas: online social platforms, corporate email communications, collaborative software development, and music recommendation systems (Table 1). These datasets differ significantly in node count, edge density, class balance, and feature sparsity, providing a comprehensive and challenging testbed for assessing the transferability, robustness, and scalability of graph learning approaches. We report performance on both node classification and link prediction tasks. For transfer learning, models are pretrained on Twitch-ES and subsequently finetuned and evaluated on Twitch-RU to measure cross-domain adaptation.

Table 1: Graph Datasets

| Dataset | #Nodes | #Edges | Avg. Deg. | Density |
|---|---|---|---|---|
| Synthetic | 9,421 | 14,949 | 3.16 | $3.4\times10^{-3}$ |
| Facebook (Rozemberczki et al., 2019) | 22,470 | 171,002 | 15.22 | $6.8\times10^{-4}$ |
| Email-EU-Core (Yin et al., 2017; Leskovec et al., 2007) | 986 | 16,687 | 33.85 | $3.4\times10^{-2}$ |
| GitHub (Rozemberczki et al., 2019) | 37,700 | 289,003 | 15.33 | $4.1\times10^{-4}$ |
| Deezer-Europe (Rozemberczki & Sarkar, 2020) | 28,281 | 92,752 | 6.56 | $2.3\times10^{-4}$ |
| Twitch-ES (Rozemberczki et al., 2019) | 4,648 | 59,382 | 25.54 | $5.5\times10^{-3}$ |
| Twitch-RU (Rozemberczki et al., 2019) | 4,385 | 37,304 | 17.02 | $3.9\times10^{-3}$ |

### 5.2 SYNTHETIC GRAPHS: STRUCTURAL SENSITIVITY ANALYSIS

To investigate the structural robustness of STRUCT-G in a controlled setting, we generate synthetic graphs using a stochastic block model (SBM) (Holland et al., 1983) augmented with triangle closure and edge rewiring operations. This setup enables precise manipulation of structural properties such as homophily and clustering coefficient while keeping other global graph characteristics fixed (e.g., diameter = 3, density = 0.01).

$$\text{Homophily} \in \{0.1,\ 0.5\} \quad \times \quad \text{Clustering} \in \{0.0,\ 0.4\}$$

producing four distinct structural regimes that test the model's ability to adapt across different levels of assortativity and local connectivity.

Each synthetic graph contains either 200, 500, of 1000 nodes with no node features, ensuring that all predictive signal must be inferred from structural topology alone. We exclusively evaluate STRUCT-G in this setting to isolate its capacity for encoding high-order structural information and to verify that its performance gains are not solely attributable to node attributes. These results are in the appendix.

### 5.3 BASELINE MODELS

To evaluate the performance, scalability, and adaptability of STRUCT-G, we benchmark against a diverse suite of graph neural network architectures encompassing classical message-passing models, structure-aware pretraining methods, and multimodal approaches. This diversity ensures a fair and comprehensive comparison across models with varying design philosophies and pretraining mechanisms. Table 2 summarizes the models included in our study. Note that all models are trained and evaluated under a unified protocol to ensure fair comparisons. We standardize training data splits,

loss functions, optimization schedules, and evaluation metrics to isolate architectural differences and minimize confounding from implementation-specific factors. This consistent experimental design allows us to quantify the relative effectiveness of structural encoding, multimodal pretraining, and task-specific fine-tuning across all competing models.

Table 2: Baseline models benchmarked in the study.

| Model | Description |
|---|---|
| **GCN** (Kipf & Welling, 2016) | Message-passing with symmetric normalization. |
| **GraphSAGE** (Hamilton et al., 2017) | Inductive aggregation over sampled neighbors. |
| **GAT** (Velickovic et al., 2017) | Neighborhood attention with learned weights. |
| **GPT-GNN** (Hu et al., 2020) (*text-augmented*) | Transformer-based GNN pretrained on textual node metadata. |
| **DeepGCN** (*naively structure-aware*) | A deep residual GNN built by us and pretrained to regress clustering coefficients. |
| **STRUCT-G** (*ours*) | Multi-task GNN built by us that combines structural and semantic signals. |
| **GraphLoRA** (Yang et al., 2025) | Parameter-efficient GNN adaptation using Low-Rank Approximation (LoRA) modules. |
| **GraphBERT** (Zhang et al., 2020) | Transformer-style architecture for graphs using attention without message passing. |
| **GPPT** (Sun et al., 2022) | Prompt-based transfer learning for GNNs that aligns pretext and downstream tasks via tokenized link prediction. |

## 5.4 TRAINING PROTOCOL, TRAINING ENVIRONMENT AND REPRODUCIBILITY

We adopt a two-stage training pipeline for applicable models: unsupervised pretraining followed by supervised fine-tuning, enabling us to assess the benefits of structural pretraining. Pretraining is applied to augmented GNN architectures such as Struct-G, GraphLoRA, GraphBert, GPPT, and DeepGCN; other baselines are trained using their standard supervised objectives. Struct-G is trained using multi-task self-supervision, combining link prediction, feature reconstruction, and alignment with Node2Vec embeddings.

All models use the Adam optimizer with a learning rate of 0.01 and weight decay of 5e-4. We train for 100 pretraining epochs and 30 fine-tuning epochs for each run. No early stopping is used. Node classification and link prediction tasks are evaluated separately, with consistent settings across all models. We use fixed random train/validation/test splits (60/20/20) generated per seed and applied uniformly across all baselines for each dataset. Node classification is evaluated using accuracy, precision, recall, F1 score, and AUC. Link prediction is evaluated using accuracy, F1 score, AUC, and average precision based on dot-product scoring over held-out edges.

We ensure full reproducibility by releasing open-source code, preprocessed datasets, and configuration files at https://anonymous.4open.science/r/adaptable-transfer-learning-47AB/readme.md.

## 6 RESULTS

### 6.1 PERFORMANCE ON DOWNSTREAM TASKS (RQ1)

The results in Table 3 show that STRUCT-G consistently achieves state-of-the-art performance across all node classification tasks, outperforming classical and pretraining-based baselines by a significant margin. This superior performance can be attributed to its hybrid architecture that fuses shallow structural priors with deep feature learning through adaptive gating. By integrating global topology information derived from random walks with node features in a task-aware manner, STRUCT-G learns expressive and generalizable embeddings that benefit from both structure and semantics.

While STRUCT-G may not achieve the absolute highest AUC on every link prediction benchmark, it consistently delivers competitive, state-of-the-art performance across diverse datasets. In cases where simpler models like GCN or GraphSAGE slightly outperform, gains are often attributable to their tendency to overfit edge neighborhoods in highly homophilic graphs. In contrast, STRUCT-G exhibits strong generalization, including on challenging synthetic datasets where node features are absent—highlighting its ability to leverage structural signals alone. These results underscore the robustness of our multi-task pretraining and structural integration approach across varied graph learning settings.

Table 3: Node classification performance (F1 Score).

| Model | Deezer | Email | Facebook | GitHub | Synthetic |
|---|---|---|---|---|---|
| Deep GCN | 0.470 ± 0.086 | 0.004 ± 0.001 | 0.436 ± 0.102 | 0.752 ± 0.029 | 0.159 ± 0.016 |
| GPT-GNN | 0.362 ± 0.005 | 0.009 ± 0.004 | 0.258 ± 0.008 | 0.425 ± 0.001 | 0.447 ± 0.010 |
| GAT | 0.612 ± 0.009 | 0.173 ± 0.037 | 0.862 ± 0.002 | 0.781 ± 0.003 | 0.562 ± 0.048 |
| GCN | 0.580 ± 0.007 | 0.220 ± 0.019 | 0.881 ± 0.004 | 0.777 ± 0.006 | 0.409 ± 0.016 |
| GraphSAGE | 0.640 ± 0.008 | 0.174 ± 0.039 | 0.879 ± 0.004 | 0.798 ± 0.006 | 0.745 ± 0.015 |
| **Struct-G** | **0.849** ± 0.027 | 0.257 ± 0.053 | **0.948** ± 0.011 | **0.851** ± 0.013 | 0.621 ± 0.020 |
| GraphLoRA | 0.560 ± 0.025 | **0.454** ± 0.071 | 0.902 ± 0.005 | 0.786 ± 0.029 | 0.357 ± 0.033 |
| GraphBERT | 0.600 ± 0.009 | 0.005 ± 0.001 | 0.860 ± 0.005 | 0.763 ± 0.014 | **0.972** ± 0.001 |
| GPPT | 0.510 | 0.380 | 0.911 | 0.773 | 0.510 |

Table 4: Link prediction performance (AUC).

| Model | Deezer | Email | Facebook | GitHub | Synthetic |
|---|---|---|---|---|---|
| Deep GCN | 0.598 ± 0.188 | 0.525 ± 0.289 | 0.772 ± 0.048 | 0.741 ± 0.330 | 0.494 ± 0.009 |
| GPT-GNN | 0.492 ± 0.012 | 0.520 ± 0.037 | 0.567 ± 0.027 | 0.500 ± 0.003 | 0.426 ± 0.013 |
| GAT | 0.813 ± 0.015 | 0.795 ± 0.034 | 0.910 ± 0.005 | 0.772 ± 0.004 | 0.533 ± 0.017 |
| GCN | **0.853** ± 0.002 | 0.709 ± 0.057 | 0.927 ± 0.002 | **0.905** ± 0.004 | 0.518 ± 0.016 |
| GraphSAGE | 0.822 ± 0.003 | **0.919** ± 0.005 | 0.911 ± 0.002 | 0.832 ± 0.006 | 0.543 ± 0.007 |
| **Struct-G** | 0.793 ± 0.023 | 0.894 ± 0.016 | **0.969** ± 0.004 | 0.894 ± 0.018 | 0.621 ± 0.020 |
| GraphLoRA | 0.816 ± 0.009 | 0.890 ± 0.007 | 0.932 ± 0.004 | 0.745 ± 0.025 | **0.691** ± 0.001 |
| GraphBERT | 0.489 ± 0.010 | 0.594 ± 0.011 | 0.860 ± 0.002 | 0.671 ± 0.027 | 0.496 ± 0.007 |

## 6.2 TRANSFER LEARNING PERFORMANCE (RQ2)

Table 5 presents the results of our transfer learning experiment. While GPPT achieves the highest classification accuracy, STRUCT-G attains the best F1 score, highlighting its robustness under class imbalance and its ability to maintain precision-recall tradeoffs. GraphBERT, on the other hand, achieves the highest AUC in link prediction, suggesting strong performance in capturing neighborhood-level similarities. These complementary strengths across models reflect the diversity of transfer learning challenges in graph domains.

## 6.3 ABLATION STUDY: OBJECTIVE-LEVEL CONTRIBUTIONS (RQ3)

Table 6 highlights the impact of removing individual self-supervised objectives from STRUCT-G on the FACEBOOK dataset. The full model (FULL) achieves the best link prediction performance (AUC = 0.976) and strong classification performance (F1 = 0.962), highlighting its effective multi-task capability. In contrast, disabling all self-supervised objectives leads to the poorest classification results. Removing either the Node2Vec-alignment or feature-reconstruction objective individually causes only a slight per-

Table 6: Impact of removing individual self-supervised losses on FACEBOOK. Higher is better. Results are single-run means; "—" denotes that the link-prediction stage was skipped.

| Variant | Node-Cls (F1) | Link-Pred (AUC) |
|---|---|---|
| FULL | 0.962 | **0.976** |
| –N2VALIGN | 0.959 | 0.971 |
| –FEATREC | **0.971** | 0.968 |
| –LINKPRED | **0.971** | — |
| –N2VALIGN AND –FEATREC | 0.946 | 0.970 |
| –CLASSIFICATION | 0.864 | 0.934 |
| –SSL (NO LP, NO ALIGN, NO FR) | 0.862 | — |
| +LINKPRED-ONLY | 0.870 | 0.935 |

formance degradations, suggesting that the model can compensate for the absence of one signal if others are active.

Interestingly, removing link prediction (–LINKPRED) does not hurt node classification and actually produces the highest F1 score (0.971), indicating that for node-level tasks, structural alignment and feature consistency may suffice. However, removing both N2VALIGN and FEATREC leads to a significant drop in classification performance (F1 = 0.946), emphasizing the complementary nature of these objectives. The weakest performance is observed when all self-supervised losses are removed (–SSL) or when only link prediction is used without supervised fine-tuning (+LINKPRED-ONLY), highlighting the necessity of multi-signal learning and task supervision for generalizable representations.

Table 5: Transfer Learning Performance: Classification (CLS) and Link Prediction (LP) Performance and Runtime

| Model | Mode | Classification Accuracy | Classification F1 | Link Prediction AUC |
|---|---|---|---|---|
| StructuralGNN | Transfer | 0.7355 | 0.4321 | 0.7414 |
| GraphBERT | Transfer | 0.7343 | 0.4234 | 0.8758 |
| GraphLoRA | Transfer | 0.7252 | 0.4204 | 0.7574 |
| GPPT | Transfer | 0.7537 | 0.4298 | — |

## 6.4 EFFECTIVENESS OF FINE-TUNING (RQ4)

Table 7 shows that task-specific fine-tuning consistently improves macro-F1 performance over structural-only pre-training across all datasets and tasks. Gains are particularly pronounced on *Email-EU-Core* and *Deezer-Europe*, where class imbalance and feature sparsity pose greater challenges. These results highlight the value of adapting pretrained embeddings to task-specific objectives, especially in low-resource or imbalanced settings. Notably, even in synthetic graphs without input features, fine-tuning yields meaningful performance gains, demonstrating the flexibility of STRUCT-G's architecture in refining representations for both node- and edge-level tasks.

Table 7: Gain in macro-F1 score (%) from task-specific fine-tuning compared with structural-only pre-training.

| Dataset | Node-Cls | Link-Pred |
|---|---|---|
| Synthetic | +6.6 | +4.8 |
| Facebook | +5.7 | +14.9 |
| Email-EU-Core | +18.9 | +28.1 |
| GitHub | +4.3 | +8.0 |
| Deezer-Europe | +24.8 | +15.5 |

## 7 CONCLUSION

We introduced STRUCT-G, a lightweight and effective framework for graph transfer learning that integrates structural priors with message-passing architectures using adaptive gating. By leveraging multi-task self-supervised pretraining which includes link prediction, feature reconstruction, and structure alignment, our method produces generalizable node embeddings suitable for diverse downstream tasks. Extensive experiments across real-world and synthetic graphs show that our method achieves strong performance in node classification and link prediction tasks while maintaining competitive runtime efficiency. These results highlight the value of explicitly modeling graph structure in pretraining without reliance on rich node attributes or external metadata.

**Limitations and Future Work.** Although STRUCT-G shows broad applicability, several limitations remain that warrant further investigation. First, inter-graph performance requires deeper analysis and enhancement to achieve stronger transfer learning outcomes. Second, additional efforts are needed to enrich the structural information captured in early-stage embeddings.

**Ethics Statement.** This work adheres to the ICLR Code of Ethics. Our study focuses on advancing graph representation learning and does not involve human subjects, personally identifiable information, or sensitive user data. All datasets used are publicly available benchmark graph datasets (e.g., Facebook, GitHub, Email-EU, Deezer), and we followed standard usage protocols as described in prior literature. We acknowledge the potential for biases that may be present in these datasets (e.g., demographic or structural skew), and our study is limited to technical benchmarking without deployment in sensitive applications.

We also disclose the use of large language models (LLMs) to support aspects of this research. Specifically, LLMs were used to assist in rephrasing text for clarity and in generating or adapting code templates. All scientific claims, analyses, and model implementations were independently validated by the authors. The use of LLMs did not influence the empirical outcomes, interpretation of results, or research integrity.

**Reproducibility Statement.** We have taken deliberate steps to ensure the reproducibility of our work. All model configurations, hyperparameters, and training details are documented in the main text and appendix. Extended experimental results, including accuracy, F1 score, AUC, and runtime

measurements, are reported with mean and standard deviation across multiple seeds. Additionally, we release complete source code, including dataset preprocessing, training pipelines, and evaluation scripts, in an open and version-controlled repository on GitHub: `https://github.com/your-repo-link` Together, these resources enable independent verification and extension of our results.

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

# Appendix

This supplementary material provides additional visualizations, structural analyses, and training details that complement the main paper. Section A presents t-SNE plots of learned node embeddings for each ablation variant of STRUCT-G, illustrating the effects of individual self-supervised objectives on the geometry of the embedding space. Section C summarizes results from a synthetic graph sweep designed to isolate the impact of key structural properties on performance. Section D details the hyperparameters used in our experiments for reproducibility.

All code, configuration files, and plotting scripts are available in our public repository at `https://github.com/programmaman/adaptable-transfer-learning`.

## A    T-SNE VISUALIZATIONS OF ABLATION VARIANTS

To qualitatively assess the representational impact of different self-supervised objectives, we visualize the learned node embeddings for each ablation variant of STRUCT-G on the Facebook dataset using t-SNE. All embeddings are taken after the final fine-tuning stage, and plotted using the same random seed and perplexity to ensure comparability. Colors indicate ground-truth class labels.

These plots help illuminate how different combinations of objectives shape the class structure in embedding space:

- **Full Model**: Contrary to expectation, the full configuration does not yield the cleanest clusters. While some peripheral classes are clearly separated, there is noticeable overlap near the center. This suggests that combining multiple objectives enhances general robustness but can introduce trade-offs that reduce fine-grained class separability.

- **No Link Prediction**: This variant produces the most visually distinct class separation. Without the link prediction objective, the model appears to rely more heavily on label supervision, forming compact and well-isolated clusters. Although this sharpens the embedding space, it reduces performance on structural tasks.

- **No Feature Reconstruction**: Cluster boundaries are present but less defined, and intra-class compactness is reduced. This indicates that feature reconstruction contributes to local coherence and tighter grouping within classes.

- **No Node2Vec Alignment**: Class regions show substantial overlap, suggesting that the alignment objective helps enforce clearer inter-class separation and more structured geometry in the embedding space.

- **No SSL (Supervised Only)**: Supervised training alone leads to clear class groupings, but the overall structure lacks nuance. The embeddings align well with labels but fail to capture broader structural variation, limiting generality.

- **Link Prediction Only**: Embeddings exhibit meaningful class separation but remain loosely distributed. This reflects that structural signals alone can support class differentiation but do not compact the space as effectively as when combined with feature-based or discriminative objectives.

Overall, these visualizations reinforce our quantitative findings: individual objectives have complementary effects. Link prediction and feature reconstruction support geometric regularity and local refinement, while supervised objectives ensure class alignment. The full model integrates these forces, though the resulting embedding space reflects a complex trade-off between separation and generalization.

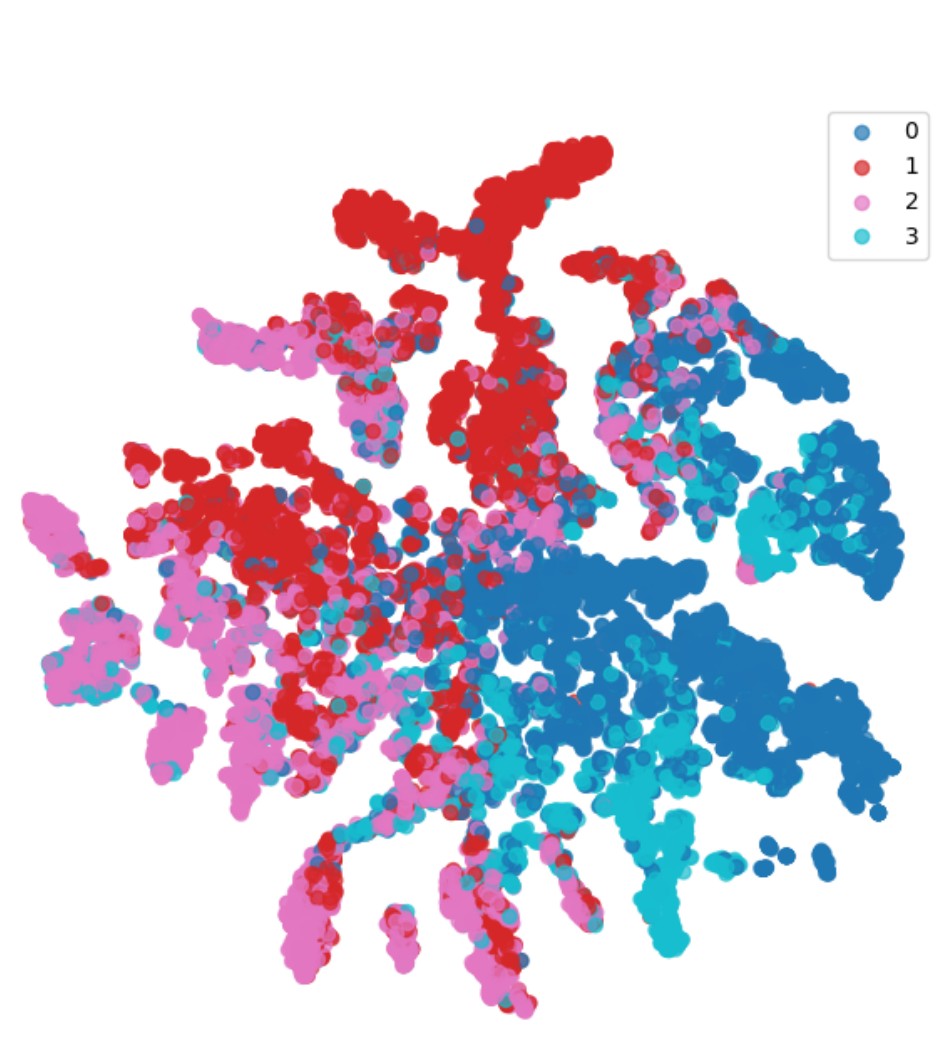

Figure 2: t-SNE embedding of STRUCT-G (Full configuration).

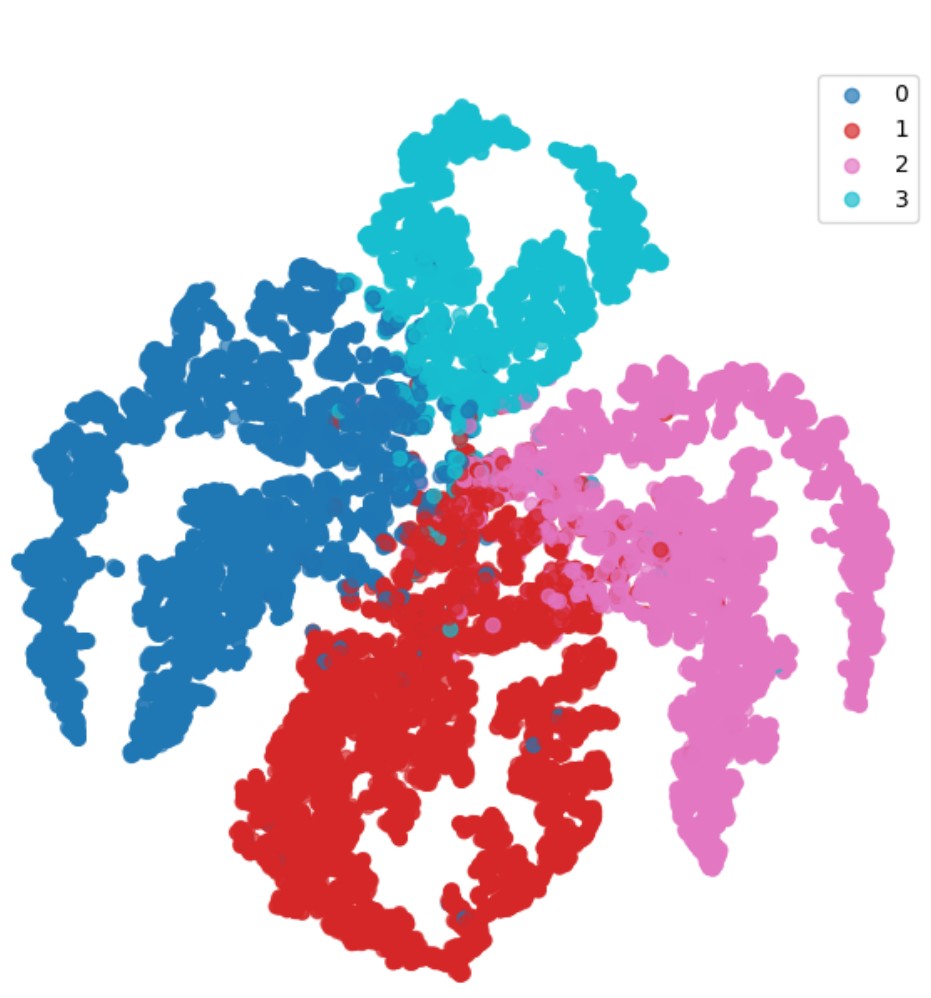

Figure 3: t-SNE embedding without the link prediction objective.

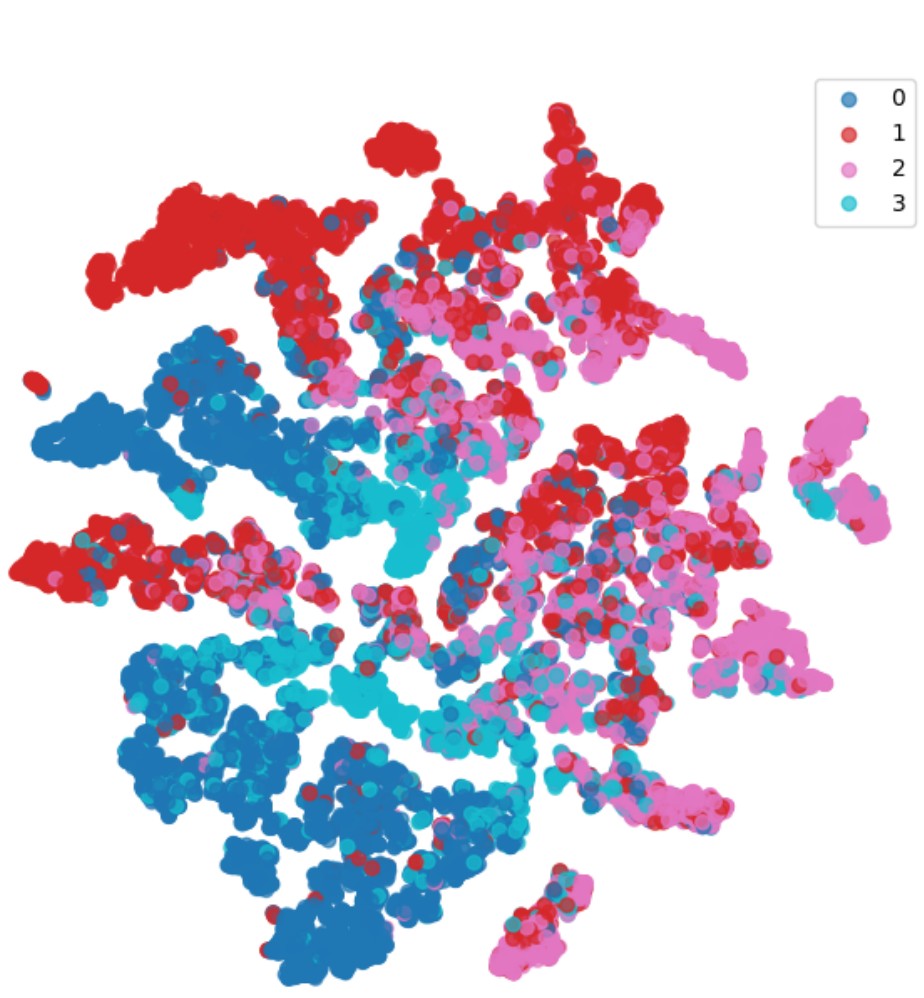

Figure 4: t-SNE embedding without the feature reconstruction objective.

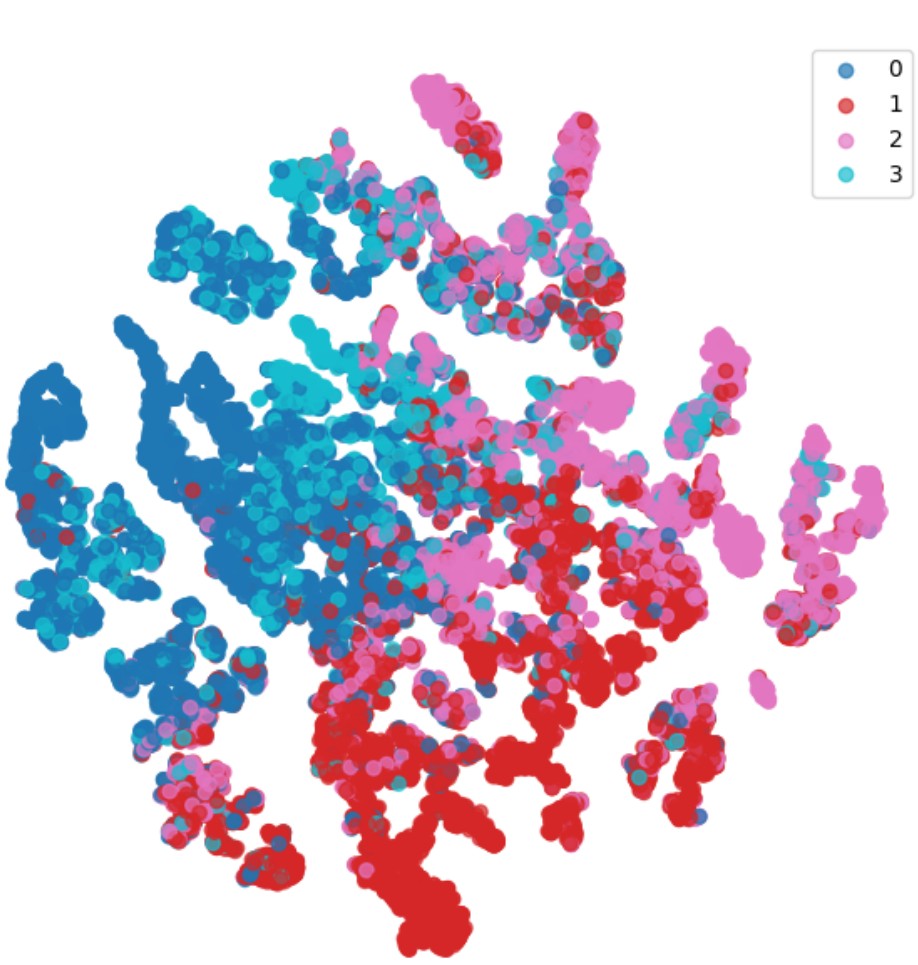

Figure 5: t-SNE embedding without the Node2Vec alignment objective.

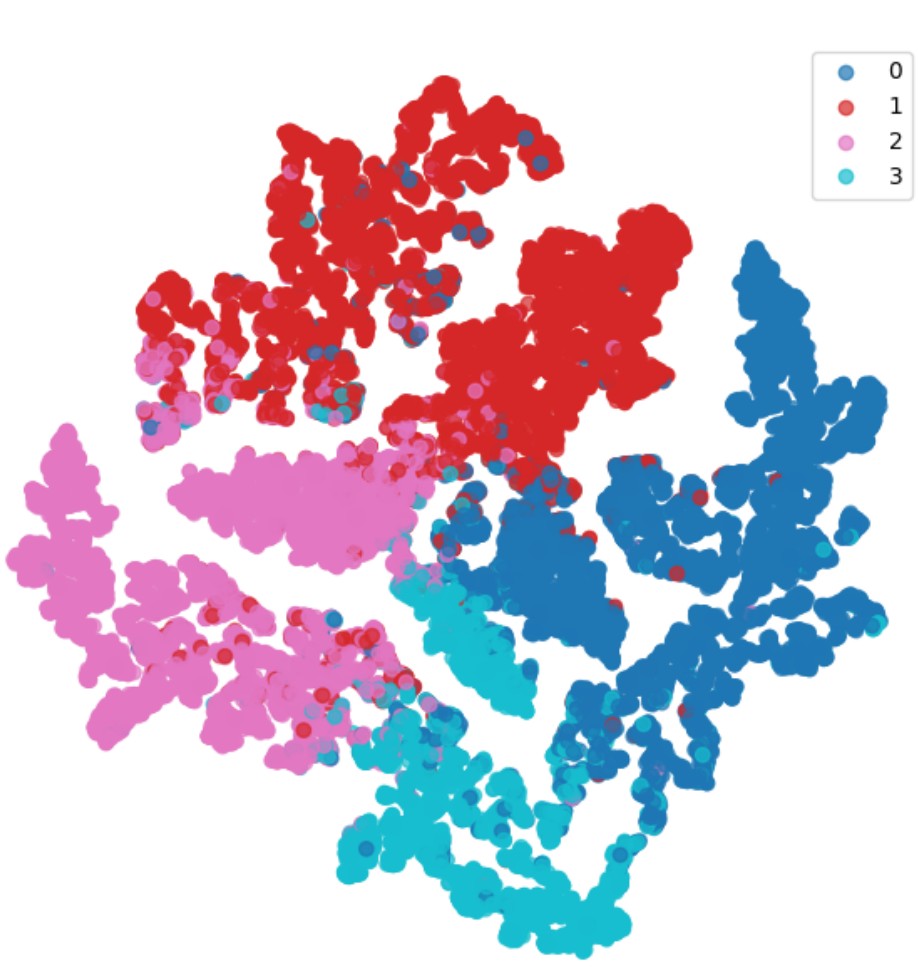

Figure 6: t-SNE embedding with no self-supervised pretraining (supervised only).

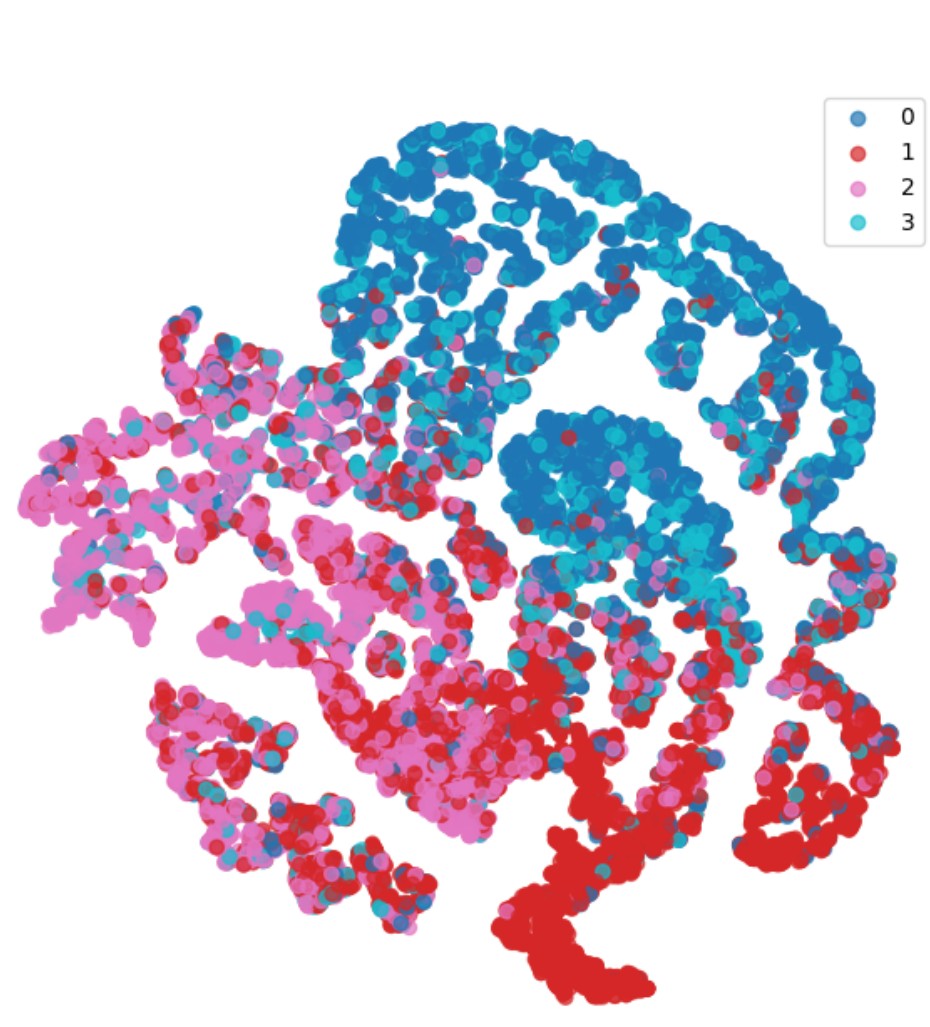

Figure 7: t-SNE embedding using only the link prediction objective.

## B  TRAINING TIME

Table 8 shows runtime comparisons for pretraining, node classification, and link prediction. While STRUCT-G is not the fastest overall, it achieves low fine-tuning times across most datasets, making it efficient for inference and task adaptation after pretraining.

An exception is the Deezer graph, where STRUCT-G has higher classification time due to the dataset's high-dimensional node features. This increases the cost of feature projection, gated fusion, and reconstruction. Overall, STRUCT-G provides a favorable trade-off: moderate pretraining cost with fast, scalable downstream performance—except in cases with unusually high feature dimensionality.

Table 8: Runtime comparison across models (in seconds). Classification and link prediction times are reported per dataset; pretraining time is averaged across datasets.

| Model | Pretrain (avg s) | Classification | | | | | Link Prediction | | | | |
|---|---|---|---|---|---|---|---|---|---|---|---|
| | | Deezer | Email | Facebook | GitHub | Synth | Deezer | Email | Facebook | GitHub | Synth |
| GNN | - | 23.42 | 1.28 | 9.94 | 15.88 | 0.83 | 23.42 | 1.28 | 9.94 | 15.88 | 0.83 |
| GAT | - | 102.85 | 4.44 | 32.16 | 55.08 | 2.42 | 102.85 | 4.44 | 32.16 | 55.08 | 2.42 |
| GraphSAGE | - | 500.26 | 12.59 | 79.35 | 130.73 | 8.28 | 500.26 | 12.59 | 79.35 | 130.73 | 8.28 |
| Deep GCN | 30.016 | 43.68 | 0.12 | 0.16 | 0.23 | 0.14 | 75.00 | 10.27 | 73.94 | 125.74 | 5.97 |
| GPT-GNN | 909.746 | 86.88 | 0.97 | 1.97 | 2.73 | 1.00 | 259.46 | 14.19 | 304.67 | 766.28 | 13.15 |
| **Struct-G** | 491.986 | 1,234.10 | 0.26 | 0.63 | 0.76 | 0.29 | 1,241.88 | 2.59 | 16.53 | 28.41 | 1.79 |
| GraphLoRA | 108.38 | 14,469.44 | 41.48 | 242.44 | 1,780.09 | 5.29 | 14,469.44 | 41.48 | 242.44 | 1,780.09 | 5.29 |
| GraphBERT | - | 26.77 | 1.98 | 26.04 | 45.37 | 9.61 | 26.77 | 1.98 | 26.04 | 45.37 | 9.61 |
| GPPT | - | 31.41 | 7.87 | 67.41 | 64.10 | 2.81 | — | — | — | — | — |

## C  PERFORMANCE TRENDS ACROSS GRAPH STRUCTURES

To further understand the influence of graph structure on learning, we performed a controlled sweep on a synthetic graph where key topological factors were varied systematically. These include homophily, clustering coefficient, assortativity, graph diameter, number of nodes, and edge factor. The analysis isolates structural effects by removing feature-based signal and measuring how STRUCT-G responds under purely topological variation.

We report the internal classification (F1) and link prediction (AUC) performance of STRUCT-G across these regimes. Results show relatively consistent trends across most dimensions, but a few structural factors exhibit more pronounced effects:

- **Homophily**: Performance is lowest at moderate homophily levels (around 0.5) and improves toward both higher and lower ends. This suggests that ambiguity in neighborhood structure may pose greater challenges than either clear homogeneity or randomness.

- **Graph Size**: Larger graphs show a mild performance drop in this synthetic regime. However, this trend likely reflects limitations of learning under random generation rather than a strict scalability issue, since STRUCT-G performs well on large real-world graphs.

- **Assortativity**: Performance peaks at neutral assortativity (around 0.0) and declines as assortative or disassortative tendencies increase. This suggests that extreme mixing patterns introduce structural irregularities that affect representation quality.

Other properties, such as clustering coefficient, diameter, and edge density, showed weak or inconsistent influence on downstream results. While these findings stem from synthetic benchmarks, they offer useful intuition for understanding which types of graphs are structurally well-suited to STRUCT-G's multi-task objectives.

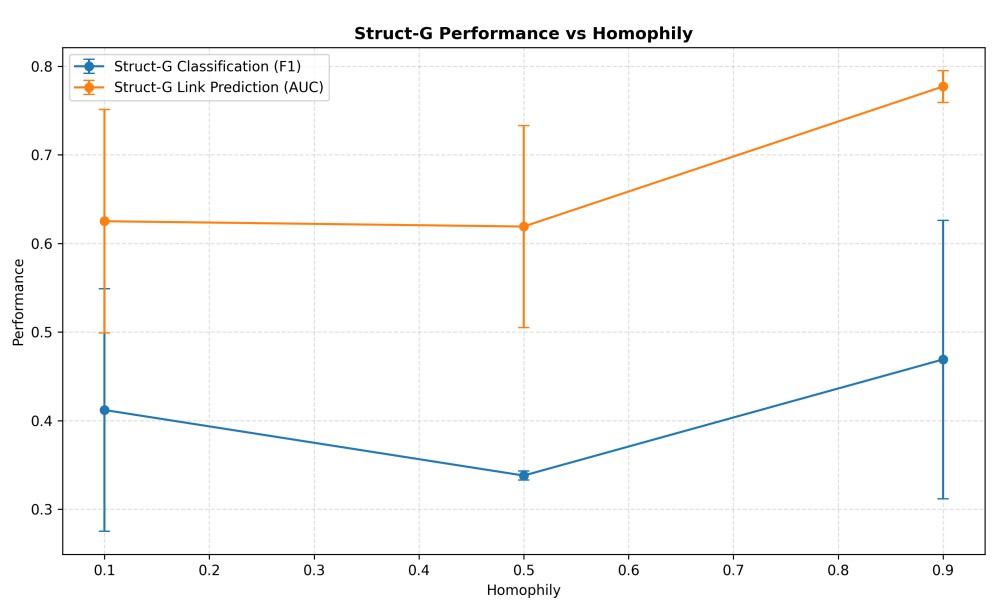

Figure 8: STRUCT-G performance across homophily values.

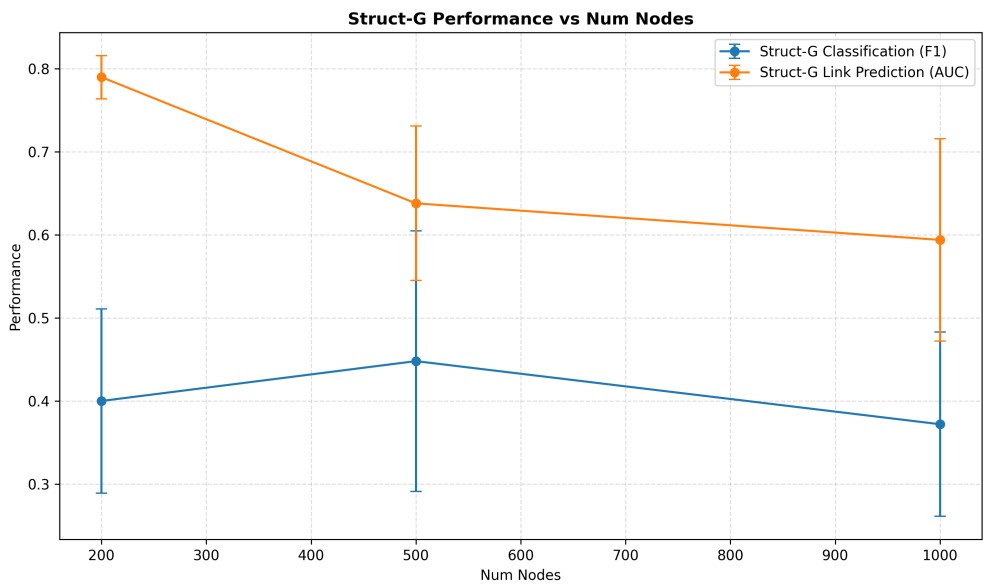

Figure 9: STRUCT-G performance across varying graph sizes.

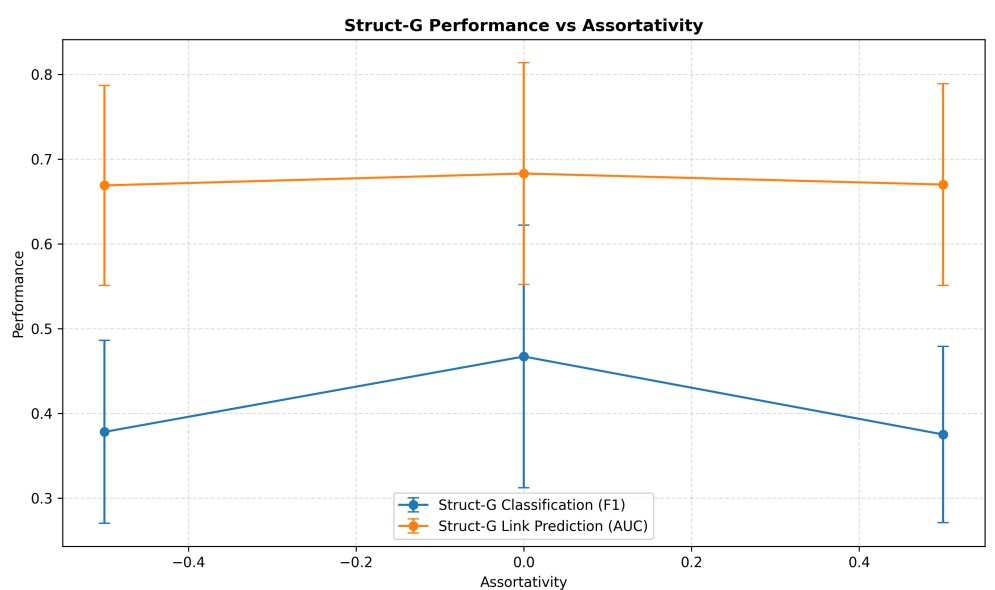

Figure 10: STRUCT-G performance across assortativity values.

# D  HYPERPARAMETERS SETTINGS

The following tables list the hyperparameters used across all experiments. Unless otherwise specified, all models were trained with the same optimizer, learning rate, and training schedule. Pretraining was used where applicable, and all results are based on fixed seeds for reproducibility.

Table 9: Hyperparameters

| Parameter | Value |
|---|---|
| Hidden dimension | 64 |
| Output dimension | 32 |
| Final embedding dimension | 128 |
| Learning rate | 0.01 |
| Weight decay | $5 \times 10^{-4}$ |
| Batch size | 128 |
| Negative samples (link prediction) | 5 |
| Pretraining epochs | 100 |
| Fine-tuning epochs | 30 |
| Train/Val/Test split | 60% / 20% / 20% |
| Optimizer | Adam |
| Loss (classification) | Cross-Entropy |
| Loss (link prediction) | Binary Cross-Entropy (logits) |
| Edge split for LP | 30% held out |

