# OpenReview forum: "Struct-G: Structural-Aware Pretraining for Graph and Task Transfer Learning"
_ICLR.cc/2026/Conference — Submitted to ICLR 2026_

### Official Review · Reviewer_pJMJ · 2025-10-30

**Soundness:** 2
**Presentation:** 2
**Contribution:** 2
**Rating:** 2
**Confidence:** 4

**Summary:**

This paper introduces, Struct-G, a framework that includes both a model architecture and a unique training objective. The model architecture uses Node2Vec to form structural embeddings, fuses them using a gated mechanism, and further processes the result through existing backbone models such as GCNs or GraphSAGE. The training objective is novel, and is basically a mixture of previously used SSL losses with an addition of a "structure-embedding <-> final feature alignment loss."
Evaluation is performed on Node Classification and Link Prediction tasks.

**Strengths:**

- Shows the effectiveness of including multiple tasks in the SSL loss
- The paper introduces a new gated operation to fuse node and position features (see weaknesses)

**Weaknesses:**

Within the use of **Node2Vec**:
- Node2Vec is NOT this paper's contribution and yet is written to be so.
- Node2Vec is just another structural embedding, and there are quite a few other options - e.g. Laplacian Eigenvectors [1], RWPE [2]. The paper makes a big deal about using Node2Vec but never ablates it with a comparison using another.

**Gating mechanism that fuses structural encoding and node features**
- The formulation presented seems more complicated than it really is. If we break $W_c$ as $W_c = [W_{x} || W_{z}]$, with $W_{x} \in \mathbb{R}^{d_h \times d_x}$ and $W_{z} \in \mathbb{R}^{d_h \times d_h}$ then the gating mechanism's equation can be simplified as:

$h^{(0)}_v = g_v \odot \hat{h}_v + (1 - g_v)\odot \check{h}_v$

$= g_v \odot (W_x x_v + W_{cz} \tilde{z}_v) + (1 - g_v) \odot W_x x_v$

$= W_x x_v + g_v \odot W_z\tilde{z}_v$

$= W_x x_v + g_v \odot W_z W_\text{proj}z_v$

Can the authors please explain why they chose to formatulate the operation in the way it is currently? I find the above simpler formulation to be easier to read. Additionally, the formulation above reveals the presence of redundant parameters. The $W_z$ and $W_\text{proj}$ are two linear layer parameters that don't need be separate. A single linear layer would be able to learn the same map with less total parameters. You can find a similar redundancy in the interaction of $W_g$ and $W_\text{proj}$ as well.
- Since this is one of the main contributions of the paper, I believe it needs a solid proof to be an improvement over simpler alternatives. The idea of fusing structural and node features is not new, people do it with simple linear layers and MLPs the concatenated $[x_v, z_v]$. Only the use of gated mechanism is new, and thus, must be ablated.

**Multi-objective Loss**
- Lack of explaination for why the feature alignment loss does not collapse. Since both $e_v$ and $z_v$ are passed through independent MLPs, what is stopping the two MLPs to collapse and output a constant vector? Can the authors confirm that the collapse does not happen?

**Evaluation**:
- Why was only one graph evaluated for transfer learning?
- The evaluation of Twitch-RU is not included in Tables 3 and 4, making it difficult to understand the impact of pretraining on Twitch-ES in Table 5.
- The ablation experiments have no measure of variance. Thus, I cannot trust any inference made from these ablations. For e.g. the improvement delta from N2Valign loss seems small, requiring an appropriate meausre of variance to see if it actually had an impact.

[1] https://arxiv.org/pdf/2012.09699

[2] https://arxiv.org/pdf/2110.07875

**Questions:**

- The multi-task objective introduces a lot of hyperparameters in the weighting of different loss components. How were these hyperparameters tuned?

---

### Official Review · Reviewer_zor5 · 2025-11-03

**Soundness:** 1
**Presentation:** 1
**Contribution:** 2
**Rating:** 2
**Confidence:** 4

**Summary:**

This paper presents STRUCT-G, a structure-aware graph pretraining framework that integrates frozen Node2Vec embeddings with an adaptive gating mechanism to combine structural and feature information. The model jointly optimizes multiple objectives—node classification, link prediction, feature reconstruction, and structural alignment—to learn transferable graph representations. Experiments on several datasets demonstrate performance comparable to standard GNN baselines.

**Strengths:**

* The paper addresses an important challenge in graph learning, transferring knowledge across graphs that differ in structure and feature space.
* The combination of node classification, link prediction, feature reconstruction, and structural alignment losses offers a unified approach to balancing supervised and self-supervised objectives.
* STRUCT-G achieves competitive or better performance than common GNN baselines across multiple datasets.

**Weaknesses:**

* The experimental section is limited, as it lacks ablations for the first two main contributions — the structural encoding step and the adaptive gating mechanism. Without these analyses, it is difficult to assess the actual impact and necessity of these components.
* The transfer experiments are restricted to a single pair of highly similar graphs (Twitch-ES → Twitch-RU), which does not convincingly demonstrate generalization. Furthermore, no results are shown for STRUCT-G trained from scratch on these datasets, making it unclear how much benefit transfer actually provides.
* The experiments are conducted on relatively uncommon graph benchmark (Facebook, Deezer, GitHub, Email-EU), which makes it difficult to compare results with prior work.
* Tables 5, 6, and 7 do not report standard deviations or variance across runs, making it difficult to draw strong or reliable conclusions from the ablation and transfer experiments.

Overall, most of the claims made by the paper are not supported by solid empirical evidence due to the lack of ablation studies, the use of non-standard benchmarks, and the absence of standard deviation reporting in the results.

**Questions:**

* I’m confused about the “lightweight structural encoding” contribution. It seems the paper just uses precomputed Node2Vec embeddings without modification. Why is this considered a key contribution rather than an implementation choice? Could the authors include an ablation comparing different structural embeddings (like laplacian)?
* Could the authors include an ablation for the proposed gating mechanism? It would be helpful to compare it with simpler fusion methods (e.g., direct concatenation or addition) to demonstrate its impact.
* The experiments are conducted on relatively uncommon datasets. Could the authors include results on more standard benchmarks such as Cora, PubMed, Amazon-CS, or Amazon-Photo to make comparisons with prior work more meaningful?

---

### Official Review · Reviewer_Tnpf · 2025-11-06

**Soundness:** 1
**Presentation:** 2
**Contribution:** 2
**Rating:** 2
**Confidence:** 4

**Summary:**

The paper introduces STRUCT-G, a pretraining framework for graph transfer learning aiming to integrate structural information into GNNs. The framework combines random-walk–based structural embeddings with raw node features through a gating mechanism and trains the model using multiple self-supervised objectives: link prediction, node classification, feature reconstruction, and structural alignment.

**Strengths:**

The paper identifies a well-motivated challenge: enabling effective transfer learning on graph-structured data and also explains why existing pretraining methods struggle with heterogeneous graph structures and sparse attributes.

The proposed feature-wise gating module is a good contribution that allows the model to dynamically balance structural and semantic information and potentially improves robustness across different graph types and downstream tasks.

The combination of multiple self-supervised objectives, such as link prediction, feature reconstruction, and structural alignment, can provide complementary learning signals that enhance representation quality and transferability.

**Weaknesses:**

- While the combination is well-motivated, many parts of STRUCT-G (e.g., random-walk embeddings, the GraphSAGE backbone, and multi-task SSL objectives) are based on known techniques. The main novelty lies in integration rather than in introducing new theoretical or algorithmic advances (not necessarily a weakness, but can be improved)

- The paper claims strong transferability across graph tasks and datasets, but cross-graph experiments (e.g., Twitch-ES –> Twitch-RU) are limited in scope, as the selected graphs are very similar. Broader evaluations across more diverse domains or unseen graph distributions would be better for the “transfer learning” claim.

- The adaptive gating mechanism is a key innovation, but the paper does not provide sufficient experimentation or interpretability analysis showing how gates behave across nodes, tasks, or structural regimes.

- While the ablation study is good regarding self-supervised objectives, it does not isolate the effects of structural embeddings or gating in depth (e.g., what happens if gating is replaced with concatenation or attention).

- The results convincingly show improved node classification and link prediction, but some other claims, such as “lightweight general foundation for graph transfer learning,” are somewhat overstated given the limited range of transfer and multi-domain tests.

**Questions:**

1- The gating module is presented as a key contribution, but its behavior is not fully explored. Can the authors include ablations or statistics (e.g., gate activation distributions, correlation with node degree or feature quality) to show how gating adapts across tasks and structural regimes?

2- STRUCT-G claims to be a lightweight foundation module. Can the authors provide clearer runtime or complexity comparisons (e.g., O-notation or empirical scaling trends) to quantify where the computational savings arise?

3- It is not clear how critical random-walk embeddings are relative to feature-based learning. Could the authors provide an ablation comparing STRUCT-G with and without frozen Node2Vec embeddings, or using alternative structural encodings (e.g., Laplacian, PPR,...)

---

### Meta-Review · Area_Chair_333R · 2025-12-22

**Summary:**

The paper proposes STRUCT-G, a pretraining framework for graph transfer learning that aims to effectively incorporate structural information into graph neural networks (GNNs). The framework leverages random-walk–based structural embeddings to capture graph structure and integrates them with raw node features through a gating mechanism. The model is jointly trained using multiple self-supervised objectives, including link prediction, node classification, feature reconstruction, and structural alignment, to learn representations with strong transferability.

The three reviewers expressed concerns about multiple aspects of the paper. However, the authors did not give any response.

**Reviewer Concerns:**

The author did not participate in rebuttal, and the reviewers also expressed their rejection.

**Reviewer Scores:**

I don't think the reviewers will change the score because the author did not participate in rebuttal.

---

### Decision · Program_Chairs · 2026-01-26

Reject